# Telomerase Activity in Somatic Tissues and Ovaries of Diploid and Triploid Rainbow Trout (*Oncorhynchus mykiss*) Females

**DOI:** 10.3390/cells12131772

**Published:** 2023-07-04

**Authors:** Ligia Panasiak, Marcin Kuciński, Piotr Hliwa, Konrad Pomianowski, Konrad Ocalewicz

**Affiliations:** 1Department of Marine Biology and Biotechnology, Faculty of Oceanography and Geography, University of Gdansk, M. Piłsudskiego 46 Av., 81-378 Gdynia, Poland; marcin.kucinski@ug.edu.pl (M.K.); konrad.ocalewicz@ug.edu.pl (K.O.); 2Department of Ichthyology and Aquaculture, University of Warmia and Mazury in Olsztyn, Warszawska St. 117, 10-719 Olsztyn, Poland; phliwa@uwm.edu.pl; 3Laboratory of Physiology of Marine Organisms, Genetics and Marine Biotechnology Department, Institute of Oceanology Polish Academy of Sciences, Powstańców Warszawy 55, 81-712 Sopot, Poland; kpomianowski@iopan.pl

**Keywords:** telomerase, triploid, rainbow trout

## Abstract

Telomerase activity has been found in the somatic tissues of rainbow trout. The enzyme is essential for maintaining telomere length but also assures homeostasis of the fish organs, playing an important role during tissue regeneration. The unique morphological and physiological characteristics of triploid rainbow trout, when compared to diploid specimens, make them a promising model for studies concerning telomerase activity. Thus, in this study, we examined the expression of the *Tert* gene in various organs of subadult and adult diploid and triploid rainbow trout females. Upregulated *Tert* mRNA transcription was observed in all the examined somatic tissues sampled from the triploid fish when compared to diploid individuals. Contrastingly, *Tert* expression in the ovaries was significantly decreased in the triploid specimens. Within the diploids, the highest expression of *Tert* was observed in the liver and in the ovaries of the subadult individuals. In the triploids, *Tert* expression was increased in the somatic tissues, while the ovaries exhibited lower activity of telomerase compared to other organs and decreased compared to the ovaries in the diploids. The ovaries of triploid individuals were underdeveloped, consisting of only a few oocytes. The lack of germ cells, which are usually characterized by high *Tert* expression, might be responsible for the decrease in telomerase activity in the triploid ovaries. The increase in *Tert* expression in triploid somatic tissues suggests that they require higher telomerase activity to cope with environmental stress and maintain internal homeostasis.

## 1. Introduction

Telomeres are non-coding regions of the genome that consist of tandemly repeated DNA sequences located at the ends of chromosomes. Telomeres not only protect coding fragments from loss during DNA replication but also safeguard chromosomes against degradation and fusion, maintain their proper topology within the nucleus, and contribute to the transcriptional silencing of genes located in the vicinity of telomeric regions [1]. Due to the “end replication problem”, telomeres shorten during each cell division, serving as a molecular clock that regulates the processes of cellular ageing and apoptosis. The loss of the telomeric tandem arrays may be compensated by telomerase, a ribonucleoprotein enzyme that consists of a catalytic subunit with reverse transcriptase activity (TERT) and an RNA template (TERC) [2]. Though telomerase expression in the somatic tissues of certain endotherms is largely absent or repressed, the enzyme activity is only present in germline cells, tumour cells, and stem cells [3,4,5]. Thus, in mammals and birds, telomeres shorten in almost all somatic cells as the organism ages [1].

In fish, it has been demonstrated that telomere shortening does not necessarily occur during ontogenesis, and its dynamics may vary depending on the species [6,7,8,9]. This phenomenon is attributed to the telomerase activity that in ectothermic organisms, including fish, is observed in the somatic cells irrespective of the organisms’ age and size [6,7,9,10,11,12,13,14] and may prevent excessive telomere shortening during the period of rapid growth [6,9].

Moreover, upregulated *Tert* expression is observed in fishes’ regenerating tissues, indicating its crucial role in the healing process of body injuries [6,11,15,16,17]. The knockout of the *Tert* gene using CRISPR/Cas9 in killifish (*Nothobranchius furzeri*) resulted in reduced fertility, atrophic testes, and ovaries, highlighting the significance of telomerase activity in maintaining organ and tissue homeostasis [18]. Research on several model fish species has demonstrated a high correlation between the expression pattern of the *Tert* gene and variation in telomerase activity, which suggests that telomerase activity in fish is primarily regulated at the transcriptional level and not at the level of protein modifications [13,14].

Rainbow trout (*Oncorhynchus mykiss*) is one of the most extensively farmed salmonid species worldwide. In 2020, its total production amounted to approximately 960,000 metric tonnes, making it the second most important species in aquaculture after Atlantic salmon (*Salmo salar*) [19]. The species is also a popular model animal in various scientific fields, including physiology, nutrition, toxicology, disease, ecology, genetics, and other fields [20,21]. Rainbow trout was the first fish species in which telomerase activity was confirmed in several organs/tissues, regardless of the fish’s age and size [10].

Spontaneous triploidization has been reported in rainbow trout, resulting from post-ovulatory oocyte ageing and the incidence of sub-lethal temperatures during fertilisation and early zygote development [22,23,24,25,26]. Additionally, artificial triploidization can be induced in this species by dispermic fertilisation of a haploid egg [27], mating tetraploid and diploid individuals [28,29,30], or by exposing eggs to chemical (colchicine, ether, deuterium oxide, and specific enzyme inhibitors) or physical (sub-lethal temperature or high hydrostatic pressure) shocks that are applied shortly after fertilisation and prevent extrusion of the second polar body and result in the development of autotriploid embryos [31,32]. The nuclear genome of autotriploid embryos is composed of haploid sets of chromosomes originating from the egg, sperm, and the second polar body [33]. The additional set of chromosomes in triploids causes cytogenetic incompatibility, impairing proper gonadal development and gamete production [34]. As a result, triploid rainbow trout females (XXX) have underdeveloped ovaries with a low number of usually aneuploid oocytes, rendering them unable to produce eggs [35]. The reproductive sterility of triploid rainbow trout females makes them an attractive subject for aquaculture production, as they are not affected by the decline in growth rate and decreased meat quality commonly observed in diploids after sexual maturation [36,37]. Moreover, sterile triploids cannot interbreed with wild fish populations if they escape from fish farms or are introduced into open waters for recreational purposes [35,38]. Triploidization has also been used as an effective sterilisation method for transgenic and potentially invasive fish species [39,40,41,42]. Triploid rainbow trout embryos have been used in reproduction studies as recipients for the interspecific transplantation of primordial germ cells to produce Atlantic salmon gametes [43]. On the other hand, it is commonly observed that triploid fish are more sensitive to sub-lethal external conditions, such as high temperatures and hypoxia, exhibiting reduced survivability and an increased incidence of skeletal malformations in comparison to their diploid counterparts [44].

Multiple studies have demonstrated that triploid rainbow trout can be an excellent model for investigating the mechanisms of gene expression regulation in autopolyploid organisms [45,46,47,48]. The unique morphological and physiological characteristics of triploid rainbow trout, including increased cell size, number of alleles/heterozygosity, sterility, continuous growth, and susceptibility to external conditions when compared to diploid specimens, make them a promising model for studies concerning telomere length dynamics and telomerase activity. Recent examinations have exhibited age-related changes in the telomeric DNA length in diploid and triploid rainbow trout females [8]. However, to the best of our knowledge, no information regarding the expression of the *Tert* gene in the somatic and reproductive tissues of polyploid vertebrates has been published to date. Therefore, the aim of this research was to explore the dynamics of telomerase expression in diploid and triploid rainbow trout females across somatic and reproductive organs.

## 2. Materials and Methods

### 2.1. Rainbow Trout Stocks, Origin and Maintenance

All-female diploid and all-female triploid stocks of rainbow trout were produced using gametes originating from breeders from a spring-spawning Rutki strain kept in the Department of Salmonid Research (DSR) of the Inland Fisheries Institute (IFI) in Olsztyn, Rutki, Poland. For this purpose, the eggs of rainbow trout were fertilised with milt from neo-males (XX) and sex-reversed genetic females. To produce triploid stock of rainbow trout, a standard protocol that involves the application of a 3-min-high hydrostatic pressure (HHP) shock (9000 psi) 35 min after egg insemination using a TRC-APV electric/hydraulic device (TRC Hydraulics Inc. in Dieppe, Canada) [34] was used. Both diploid and triploid rainbow trout individuals were reared separately under the same husbandry in the hatchery plastic tanks (1 m^3^) (first year) and outdoors in the rectangular (10 m^3^) (second year) and circular concrete ponds (56 m^3^) (third year of rearing). The fish were fed daily, and feeding rates were adjusted to their growth and diurnal temperature. Detailed information on environmental conditions during fish rearing has been described by Panasiak et al. (2020) [8]. Diploid and triploid females within the second (subadults) and third (adults) years of life were randomly chosen and sacrificed by an overdose of MS-222 (Appendix A). For examination of telomerase expression and the histopathology of ovaries, five specimens from each age and ploidy stock were sampled. Cytogenetic analysis was applied for ploidy confirmation.

### 2.2. RNA Extraction and Analysis of Rainbow Trout Telomerase (Tert) Expression

The liver, spleen, muscles, gills, and ovaries were sampled and immediately submerged in RNAlater (Thermo Fisher Scientific, Waltham, MA, USA). Preserved samples were kept at −20 °C until further analysis. Total RNA was extracted using the Bead-Beat Total RNA Mini kit (A&A Biotechnology, Gdańsk, Poland), following the manufacturer’s instructions. Residual DNA in the extracted RNA samples was removed using the Clean-Up Concentrator kit (A&A Biotechnology, Gdańsk, Poland). RNA concentration and purity were measured with a NanoDrop One spectrophotometer (Thermo Fisher Scientific, Waltham, MA, USA), and RNA integrity was assessed by 1% agarose gel electrophoresis. The obtained RNA samples were immediately processed further.

Purified total RNA samples of satisfactory quality were used to synthesise cDNA using the RevertAid First Strand cDNA Synthesis Kit (Thermo Fisher Scientific, Waltham, MA, USA). The reaction mixtures were prepared in a total volume of 20 µL, composed of 1X Reaction Buffer, 5 µM of Random Hexamer primer, 1 µM of dNTP Mix, 20U of RiboLock RNase Inhibitor, 20U of RevertAid M-MuLV RT reverse transcriptase, and 1 µg of RNA sample. The reverse transcription reactions were carried out on a Mastercycler^®^ X50a (Epppendorf, Germany). The samples were incubated for 5 min at 25 °C, followed by 60 min at 42 °C, and terminated by heating at 70 °C for 5 min. The obtained cDNA samples were diluted 1:10 in DEPC-treated water and stored at −20 °C for further analysis.

The primer sequences for the *Tert* gene were designed based on genetic information deposited in the GenBank (accession numbers: KC204724-32, XR_005041402, HM852030, release 101) and Ensembl (accession numbers: ENSOMYG00000034782, release 109) databases. All available isoforms and splice variants of the *Tert* gene were taken into consideration before primer construction. The primers were designed with the Primer Blast designing tool (NCBI) with default parameters, including the primer location on the exon-exon junction and a qPCR product length of approximately 100 bp.

Real-time PCR analysis was performed using designed primers for the *Tert* gene (Forward: 5′-CTCTTCCATCACCCCTGCTC-3′, Reverse: 5′-CCCCACTCACATCCACCTTG-3′). β-actin (*Actb*) was chosen as the housekeeping gene due to its proven stable expression level across different tissues (Forward: 5′-GCCGGCCGCGACCTCACAGACTAC-3′, Reverse: 5′-CGGCCGTGGTGGTGAAGCTGTAGC-3′) [49]. The qPCR was carried out separately for housekeeping and target genes by a Cielo 6 Real-Time PCR System (Azure Biosystems, Italy) using the PowerTrack SYBR Green Master Mix (Applied Biosystems, California, USA). Reaction efficiencies were estimated from the slopes of the standard curves made of 5-point, and 10-fold serial cDNA dilutions starting from 10 ng/uL. The optimal reaction conditions for both assays displayed an efficiency between 99 and 101%. The qPCR reaction mixtures were prepared in a total volume of 10 µL, consisting of 1X PowerTrack SYBR Green Master Mix, 0.5 µM (*Tert*) and 0.15 µM (*Actb*) of each primer, and 1 µL of cDNA. The Real-Time PCRs were run in triplicates with the following thermal cycling conditions: an initial polymerase activation step at 95 °C for 5 min, followed by 35 cycles of 95 °C for 30 s (denaturation), 20 s at 62 °C (primer annealing), and 72 °C for 15 s (elongation). During each run, negative controls using pure water and non-transcribed RNA were used to exclude contamination. The analysis of the melting curve (60–95 °C) at the end of each run concluded the protocol. Fluorescence data were collected after the elongation step and in 0.1 °C steps on the melting curve. No splice variants were observed in the analysed organs. The relative expression was calculated based on the difference between Ct values for reference and target genes using the Livak and Schmittgen equation [50].

### 2.3. Histological Preparation

Ovaries were removed and fixed with Bouin’s solution. After preservation in Bouin’s solution, fragments of the ovary tissue samples were dehydrated in 70% ethyl alcohol, treated with xylene, and submerged in paraffin blocks. Slices 4–5 µm thick were cut using a LEICA RM 2165 rotational microtome (LEICA Microsystems, Wetzlar, Germany) and stained with the HE (haematoxylin-eosin) topographic method [51]. Histological analyses of cross-sections to determine the shape, size, and type of germ cells present in the gonads were performed using a LEICA DM 3000 transmission light microscope and the micro image computer analysis software LEICA QWin Pro (LEICA Microsystems AG, Heerbrugg, Switzerland). The nomenclature of cellular structures and germ cells in the analysed gonads was adapted according to Brown-Peterson et al. (2011) [52].

### 2.4. Ploidy Confirmation: Preparation of the Metaphase Spreads and Microscopic Analysis

Somatic metaphase chromosomes were prepared from the cells of a cephalic kidney, a primary fish lymphoid organ. Portions of the cephalic kidney were removed, placed in tubes with 5 mL of KCl (0.075 M) and 10 µL of 0.3% colchicine (Sigma-Aldrich, St. Louis, MO, USA) solution, and incubated for 20 min. at room temperature. Fragments of tissues were then macerated with scissors, dissociated by pipetting to obtain homogenous cell suspensions, and left for hypnotization for another 45 min. Next, 10 drops of freshly prepared ice-cold fixative (methanol: acetic acid, 3:1) were added to the tubes with cell suspensions. After 1 min, the tubes were filled with the fixative up to 8 mL and centrifuged at 160× *g* for 10 min. Then, the supernatant was tossed out, and the cell pellet was resuspended in freshly prepared fixative. Samples were kept at −20 °C for 30 min. The fixative was changed three times. After the final centrifugation, the supernatant was replaced by a freshly prepared fixative (1–2 mL). Microscope slides were prepared by placing one or two drops of the cell suspension from a height of about 30 cm onto a clean slide and leaving it to dry.

The slides with the chromosomes were mounted using antifade solution Vectashield containing DAPI (4′,6-diamidino-2-phenylindole) (1.5 μg/mL) (Vector, Burlingame, CA, USA) and analysed using the BX53 Olympus microscope (Olympus, Japan) equipped with epifluorescence, an appropriate filter set, and a dedicated 5 M CMOS camera (Applied Spectral Imaging, Migdal Haemeqck, Israel). Images of the metaphase chromosomes were captured, and the electronic processing of the images was performed using GenASIs 8.1.1 software (Applied Spectral Imaging, Israel).

### 2.5. Statistical Analysis

The fold change expression values of *Tert* relative to the reference gene were analysed using R software version 2022.12.0 (20 February 2023). The normal distribution was assessed using the Shapiro-Wilk test, and the homogeneity of variances between the analysed groups of rainbow trout was determined using Levene’s test. The student t-test, or Kruskal-Wallis’ test, was used, based on the data distribution, to determine significant differences (*p* < 0.05) in the telomerase expression between the organs of the fish from the examined groups.

## 3. Results

### 3.1. Confirmation of Ploidy—Cytogenetic Examination

The chromosome number in rainbow trout from the diploid stock varied from 59 to 63, while fish from the triploid stock exhibited 89 to 94 chromosomes. The variation in the chromosome numbers observed here originated from Robertsonian translocation, which in rainbow trout is a common characteristic [53,54]. The differences observed in the number of chromosomes showed that the studied fish differed in terms of their ploidy.

### 3.2. Histology of Gonads

Macroscopically, the gonads of the diploids were composed of paired, correctly shaped, similar-sized lobes located cephalically in the peritoneal cavity, embedded just below the wall of the swim bladder (Figure 1A). In contrast, in the triploid fish, these structures were characterized by extremely thin, poorly demarcated bands of tissue adhering parallel along the ventral port of the swim bladder wall (Figure 1B).

Histological analysis of the gonads of the diploids revealed the presence of only ovaries filled with oocytes at the previtellogenesis stage, accompanied by single oogonia nests. Some of the fish were characterized by the presence of distinct ovarian lamellae, typical of salmonid ovaries, within the structure of the gonads (Figure 2A), while others lacked any elements separating groups of previtellogenic oocytes (Figure 2B). In the subadult fish, the presence of a small number of vitellogenic oocytes at the initial stage of vacuolization was noted (Figure 2C). In the ovaries of the triploid individuals, two types of ovarian tissue structures were observed on the histological sections. In some examined specimens, the interior of the organ structure consisted exclusively of the somatic connective tissue cells (fibrocytes) and surrounding fibrous elements enclosed in a connective tissue structure surrounded by several layers of differentiated epithelial cells. In the cross sections of such ovaries, faintly outlined structures resembling ovarian lamellae were observed (Figure 2D). In other triploids, the structure of the gonads was disrupted, and the ovaries were filled with connective tissue elements with single (Figure 2E) or few previtellogenic oocytes (Figure 2F) located in the central part of such gonads.

### 3.3. Expression of the Tert Gene

Real-time PCR was used to quantify the *Tert* mRNAs, which were detected in all the analysed organs. Generally, significantly (*p* < 0.05) increased expression of the *Tert* gene was observed in all the examined somatic tissues from triploid specimens when compared to diploids. A contrasting situation was reported in the ovaries sampled from triploid individuals, which showed decreased expression (Figure 3). Upregulated *Tert* expression was detected in muscles sampled from adult (3-year-old) rainbow trout compared to subadult (2-year-old) fish; however, the observed differences were significant (*p* < 0.05) only among diploids. In the gills, no significant differences in *Tert* expression were reported between subadult and adult fish. In the liver, significantly (*p* < 0.05) higher *Tert* expression in triploids compared to diploids was detected only in the adult trout, while transcription levels of the *Tert* gene in the subadults were significantly lower in the triploids than in the diploids (Figure 3). The *Tert* expression level in the spleen was essentially increased in adult triploids when compared to all other fish groups. No differences in *Tert* expression levels were observed between subadult diploids, triploids, and adult diploid trout. In the ovaries, the transcription level of *Tert* mRNAs in diploids was significantly higher (*p* < 0.05) than in triploids, regardless of the age of the fish. Additionally, subadult individuals exhibited a slightly higher expression of *Tert* than adult specimens in both the diploid and the triploid groups, though the differences were not significant (*p* < 0.05).

## 4. Discussion

Compared to their diploid counterparts, triploids exhibit three fundamental differences: they are more heterozygous, they have fewer larger cells in the tissues, and their gonadal development is disrupted [34,55]. Despite the commonly held belief that larger cells have lower metabolic rates per unit of mass [56], the reported energy metabolic and oxygen consumption rates in triploid fish are usually similar or even higher than those in diploid individuals [34,55]. Likewise, the developmental rates of triploids are generally comparable to those of diploids, although in some cases they may be slightly lower or higher [34,55]. However, after the sexual maturation of diploid fish, triploids tend to exhibit faster growth as they do not invest energy in gonadal development [33]. Triploids also display a higher ratio of malformation incidences compared to diploids, which is based on their genetic perturbations resulting from the ploidy change, the detrimental effects of physical shock applied for triploidization, and distinct nutritional requirements [57,58]. A depleted resource transport capacity (especially of oxygen) in larger cells, attributed to longer diffusion distances and reduced membrane surfaces, limits triploids’ aerobic energy budget and abilities to accumulate energy reserves [59]. Triploids may also display an increased mortality rate as they are less resistant to chronic hypoxia and high temperatures than diploids. Nevertheless, under optimal conditions, both triploids and diploids actually tend to display similar mortality rates [34,55].

Despite the large cellular architecture and physiological differences between triploids and diploids, the effect of triploidy on gene expression is surprisingly smaller than expected, as triploid and diploid fish have usually comparable levels of expression in their somatic tissues (a dosage compensation phenomenon) [60,61,62,63,64,65]. Only a small number of genes may be subjected to both positive and negative dosage compensation in triploid fish [66]. Most of the up- or down-regulated genes reported so far in the triploids are those regulating metabolism and stress responses, which is hypothesised to be associated with the higher susceptibility of triploids to hypoxia and thermal and oxidative stresses when compared to diploids [34,55,63,64].

Research on *Tert-/-* knock-out zebrafish mutants revealed severe histopathological abnormalities in their testes, liver, intestine, gills, pancreas, kidney, and muscle tissues, which suggests that the activity of telomerase in fish, irrespective of their age, plays an important role in the maintenance of tissue homeostasis [67]. Thus, cellular and physiological differences between diploid and triploid fish, including a larger size of cells, a potentially changed standard metabolic rate, and decreased resistance to demanding conditions, may require increased telomerase activity in triploids to sustain the proper physiology of their tissues. Rearing conditions optimal for the diploid rainbow trout examined here might have been more demanding for the triploid individuals, so the maintenance of tissue homeostasis required increased telomerase expression.

Telomerase expression and activity have been extensively studied in the muscles of various fish species [9,10,17,68,69]. However, the precise function of telomerase during somatic growth and its connection with the indeterminate growth of fish remain unclear. For instance, a study on the European hake (*Merluccius merluccius*) revealed a positive correlation between the levels of *Tert* transcription and body size, while opposite findings were reported in the Atlantic cod (*Gadus morhua*) and the rainbow trout [10,70]. Frequently reported up-regulation of telomerase transcription and activity during early developmental stages in fish suggests its role in promoting tissue cellular proliferation [9,10,11,14,70]. Nevertheless, in our previous study, differences in telomerase activity in muscles from trout siblings showing normal or retarded growth were insignificant [71]. Differences in the expression of *Tert* in the muscles of subadult and adult diploid and triploid rainbow trout may reflect a disparity in energy expenditure between fertile and sterile individuals for somatic growth. Triploid salmonid females do not sexually mature, so they do not need to spend as much energy on their gonadal development and gamete production as diploids do. However, elevated rates of hypertrophic growth in triploids, which possibly demand greater energy expenditure, might be considered to explain the up-regulated expression of *Tert* in the muscles of triploid fish [55,72].

The liver, spleen, and gills are organs in fish that are known for their remarkable regenerative capabilities and resistance to oxidative stress [6,11,69]. The ability of these organs to regrow after injury or damage has been linked to increased telomerase activity [7,11,15,16,17]. Triploid rainbow trout have been found to be more resistant to carcinogenesis and have increased fin regenerative abilities when compared to diploid fish [73]. Our study’s findings indicate that the increased regenerative abilities of triploids may be attributed, among others, to the upregulation of telomerase expression in their somatic tissues [35,74]. Moreover, the increased telomerase activity observed in triploid rainbow trout might explain the higher resistance of the polyploid cells/tissues to oxidative stress [75]. The liver serves as a vital organ for regulating organisms’ energy metabolism, but it is also highly exposed to the toxic compounds and reactive oxygen species (ROS) that arise during metabolic processes [76]. In fish, the liver is the organ with the highest level of telomerase expression and activity, which is thought to be linked with the ability of this organ to cope with oxidative stress [10,71]. The significantly upregulated *Tert* expression in the liver of the subadult diploid fish when compared to the triploids might reflect a surge in the metabolic rate associated with the sexual maturation process and vitellogenin synthesis, among others. The decreased expression of *Tert* observed in adult fish may be correlated with a lower liver metabolic rate as the process of gonadal maturation in triploids is accomplished. Nevertheless, further research is needed to explain the background of this phenomenon. Multiple studies have demonstrated that triploids require higher gill irrigation rates to compensate for their lower respiratory efficiency and oxygen supply compared to diploids [54,77]. The higher rate of opercular movement in triploids can potentially increase their exposure to environmental stressors; thus, up-regulated *Tert* transcription reported in the gills of triploid trout might play some role in maintaining their homeostasis and regenerative abilities. In turn, the significantly increased levels of telomerase expression in the spleen of adult triploid rainbow trout observed here may be linked with the increased synthesis and accumulation of erythrocytes observed in the triploid rainbow trout, which was thought to be a compensatory response to a lower oxygen supply [78].

Telomerase expression in mammalian adults is restricted to highly proliferating cells, including female germline stem cells that can differentiate into oocytes. Moreover, telomerase activity has been confirmed in the oocytes at different stages of development [79,80] and in the ovarian granulosa cells that proliferate in the developing follicle and form a single layer around the oocytes [81,82,83]. In fish, expression of *Tert* varies significantly between tissues, and gonads are the organs with the highest telomerase activity [6,84,85]. This is also observed in the diploid rainbow trout examined here. Telomerase plays an important role during fish ovarian development, egg production, and maturation. *Tert*-deficient fish are characterized by atrophied ovaries, reduced egg production and premature infertility [18,67]. A dramatic decrease in telomerase activity has also been observed in the ovaries in triploid rainbow trout that are considered sterile. The ovaries in triploid rainbow trout are highly reduced, with only a few primordial germ cells and several, usually euploid or aneuploid, oocytes [34,86,87,88]. The gonads of the triploid trout studied here were actually devoid of oocytes and consisted exclusively of somatic connective tissue cells, mostly fibrocytes (Figure 2). The sterility in triploid rainbow trout females results from the arrest of oogonium development, failure of meiosis, and lack of interactions between oocytes and follicular cells. Moreover, in triploid ovaries, the development of follicular cells is inhibited and production of estradiol decreases, which in turn causes a decline in the synthesis and secretion of hepatic vitellogenin (VTG) [89], a major precursor of the fish egg yolk protein that is indispensable for oocyte growth and maturation in fish. Moreover, oocytes in triploid rainbow trout may undergo apoptosis [90]. The presence of only a few germ cells and oocytes that are usually characterized by high telomerase activity may therefore be responsible for the reduced expression of *Tert* observed in the ovaries from triploid rainbow trout females. Furthermore, it has been confirmed that telomerase is activated by oestrogen through the stimulation of *Tert* expression [91], hence a deficiency of estradiol observed in the ovaries of triploid rainbow trout [92] may also be responsible for the decreased expression of *Tert*.

## 5. Conclusions

Ploidy compensation is known as an important process that ensures the equal distribution of protein production across different ploidy levels in organisms. Recent research on dosage compensation in polyploid fish has shown that diploid and triploid individuals exhibit similar levels of gene expression in most analysed tissues, indicating compensation mechanisms are in place. However, some genes may show either positive or negative dosage compensation in triploid fish, with most up- or down-regulated genes in triploid salmonids being related to metabolism and stress response. The positive dosage compensation of the *Tert* gene expression in several organs of triploid rainbow trout females found in our study supports the notion that telomerase activity is essential in sustaining fish tissue homeostasis/biology. This study adds to our understanding of the relationship between ploidy and telomerase expression in fish and suggests that telomerase may play a more significant role in the biology of triploid organisms than previously thought. The significant downregulation of *Tert* expression in the ovaries of triploid rainbow trout, as observed in our study, appears to be correlated with their atrophied state, decreased egg production, and infertility. The absence of oocytes in triploid ovaries, along with their inhibited follicular cell development, might contribute to the recorded decline in telomerase expression. Nevertheless, further research is needed to explain the different patterns of telomerase expression between diploids and triploids across different organs reported here, which will allow us to better understand the relationship between ploidy and *Tert* gene expression in vertebrates.

## Figures and Tables

**Figure 1 cells-12-01772-f001:**
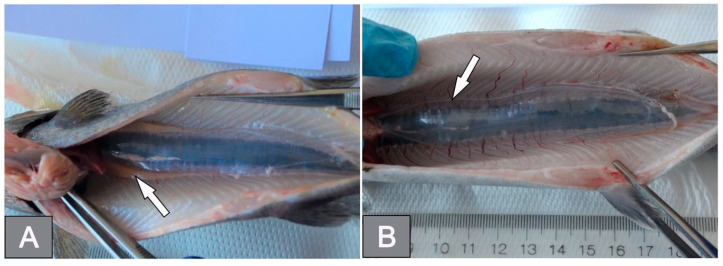
Macroscopic structure of the rainbow trout gonads: (**A**) diploid female; (**B**) triploid female.

**Figure 2 cells-12-01772-f002:**
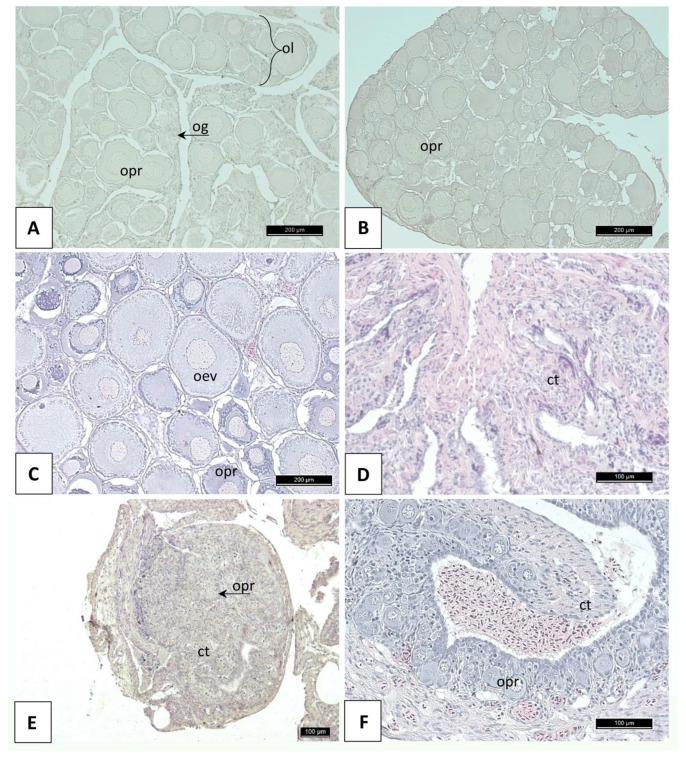
Histological cross-sections of the rainbow trout gonads: (**A**) the ovary of a diploid with typical ovarian lamellae; (**B**) the ovary of a diploid individual formed of ovarian lamellae; (**C**) the ovary of a diploid fish; (**D**) the sterile ovary-like organ of an individual filled with connective tissue cells and fibrous elements; (**E**) the ovary-like gonad of a triploid individual with a single previtellogenic oocyte located in the central part of the organ; (**F**) the ovary-like gonad of a triploid individual with a few previtellogenic oocytes. Description: ct—connective tissue elements; og—oogonia; oev—early vitellogenesis oocyte; ol—ovarian lamina; opr—previtellogenic oocyte.

**Figure 3 cells-12-01772-f003:**
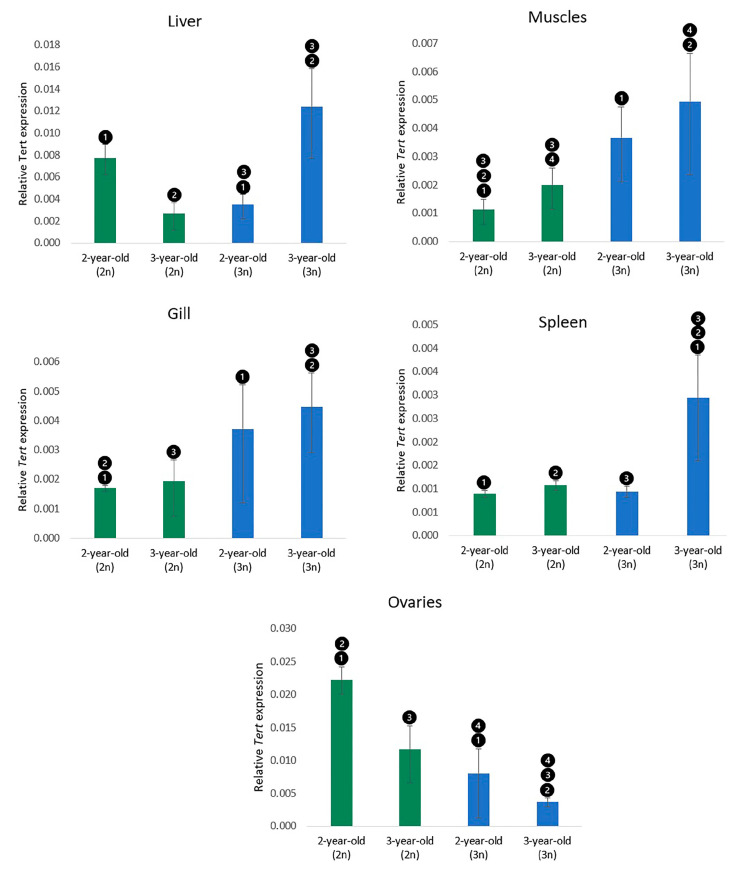
Relative *Tert* expression in the organs of rainbow trout. The same numbers indicate statistically significant (*p* < 0.05) differences between groups. Values are presented as fold changes relative to the reference gene (*β-actin*). The measure of variation is derived from the respective SEM of the Ct values.

## Data Availability

The data presented in the current manuscript are available from the corresponding author upon request.

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
