# Peer review of "Telomerase Activity in Somatic Tissues and Ovaries of Diploid and Triploid Rainbow Trout (Oncorhynchus mykiss) Females"

_cells, 2023, doi:10.3390/cells12131772_

Round 1

Reviewer 1 Report

This manuscript examined the expression of the Tert gene in various organs of sub-adult and adult diploid and triploid rainbow trout females. The results revealed that upregulated Tert mRNA transcription was observed in all the examined somatic tissues sampled from the triploid fish. However, there are some improvements can be made and confusion needs to be explained in this manuscript before it could be considered to accept.

1. In the introduction section, if “telomere shortening does not necessarily occur during ontogenesis”, is the function of telomerase also necessary in fish?

2. In this study, only the telomerase (Tert) mRNA expressions were detected, however, the protein and enzyme activity expression also should be detected.

3. Is only telomerase involved in the synthesis telomeric DNA? Are there other genes that are also involved in the synthesis process? The related gene expressions also should be detected.

4. Line 21: Reword “was” to “were”

5. Line 288: Delete the word “a” after “both”

6. Line 358: Reword “of” to “in” after “decrease”

7. Line 360: Add “a” after “only”

8. Line 372: Reword “from” to “of” after “ovaries

9. Figure 3: The resolution is not enough, please use a figure with a resolution greater than 300ppi.

Minor editing of English language required.

Author Response

Response to Reviewer 1

Remark 1: In the introduction section, if “telomere shortening does not necessarily occur during ontogenesis”, is the function of telomerase also necessary in fish?

Response: In some fish species, erosion of telomeres  with age does not occur and its dynamics may vary depending on the species what is attributed to activity of telomerase. Thus, one of the function of telomerase in the fish somatic tissues is preventing telomere shortening. Additional information about telomerase function has been included in the introduction. (See lines 59-62, 37-45)

Remark 2: In this study, only the telomerase (Tert) mRNA expressions were detected, however, the protein and enzyme activity expression also should be detected.

Response: In fish examined to this regard, pattern of Tert gene expression is highly correlated with telomerase activity so regulation of the Tert gene transcription is one of the fundamental mechanisms regulating the activity of the enzyme. Additional information has been included in the introduction. (See lines 65-68)

Remark 3: Is only telomerase involved in the synthesis telomeric DNA? Are there other genes that are also involved in the synthesis process? The related gene expressions also should be detected.

Response: Telomeric sequences are synthesized by the telomerase reverse transcriptase, and, as such, telomerase plays a very important role in the synthesis of telomeric DNA. In fish no other mechanism responsible for telomere elongation has been found to date. Information has been included in the introduction.

Remark 4-8: Remarks on grammar corrections

Response: We applied all the changes. (see changes as the mode track changes has been ON)

Remark 9: The resolution is not enough, please use a figure with a resolution greater than 300ppi.

Response: We changed the figure and use the requested resolution.

Reviewer 2 Report

Telomerase activity in somatic tissues and ovaries of diploid and triploid rainbow trout (Oncorhynchus mykiss) females by Panasiak et al.

 Provided that triploid rainbow trout has been proven to be an adequate model for investigating the mechanisms of gene expression, Panasiak et al. aimed to explore the dynamics of telomerase expression in diploid and triploid rainbow trout females across somatic and reproductive organs. For that purpose, diploid and triploid females were used for examination of telomerase expression and the histopathology of ovaries i.e., RNA extraction and analysis of telomerase (Tert) expression, histological preparation to characterize germ cells and cytogenetic analysis was applied for confirmation of the ploidy.

 The manuscript is beautifully written, and the good illustrations support the claimed results:  authors hypothesize that the maintenance of the tissue homeostasis required increased telomerase expression in triploids. Despite dosage compensation studies in fishes determine that diploid and triploid individuals exhibit similar levels of gene expression in somatic tissues, up and down regulation of some genes metabolically related are described in triploid fish. The authors clearly show that Telomerase activity is essential in sustaining fish tissue homeostasis/biology, e.g., the downregulation of Tert expression in the ovaries of triploid rainbow trout appears to be correlated with their atrophied state, decreased egg production, and infertility.

Congratulations for your excellent work.

Author Response

We would like to thank the Reviewer for taking the time to review our manuscript. We appreciate your thorough evaluation of our work and your positive feedback. We are glad to hear that our research has been well received and that no corrections or revisions are necessary. Your positive feedback reinforces our confidence in the quality and significance of our findings.